# Vegetation recovery in tidal marshes reveals critical slowing down under increased inundation

Jim van Belzen[1], Johan van de Koppel[1,2], Matthew L. Kirwan[3], Daphne van der Wal[1], Peter M.J. Herman[1,4], Vasilis Dakos[5,6], Sonia Kéfi[6], Marten Scheffer[7], Glenn R. Guntenspergen[8] & Tjeerd J. Bouma[1,2]

A declining rate of recovery following disturbance has been proposed as an important early warning for impending tipping points in complex systems. Despite extensive theoretical and laboratory studies, this 'critical slowing down' remains largely untested in the complex settings of real-world ecosystems. Here, we provide both observational and experimental support of critical slowing down along natural stress gradients in tidal marsh ecosystems. Time series of aerial images of European marsh development reveal a consistent lengthening of recovery time as inundation stress increases. We corroborate this finding with transplantation experiments in European and North American tidal marshes. In particular, our results emphasize the power of direct observational or experimental measures of recovery over indirect statistical signatures, such as spatial variance or autocorrelation. Our results indicate that the phenomenon of critical slowing down can provide a powerful tool to probe the resilience of natural ecosystems.

[1] Department of Estuarine and Delta Systems, Royal Netherlands Institute for Sea Research (NIOZ) and Utrecht University, PO Box 140, Yerseke NL-4400 AC, The Netherlands. [2] Groningen Institute for Evolutionary Life Sciences, University of Groningen, PO Box 11103, Groningen 9700 CC, The Netherlands. [3] Virginia Institute of Marine Science, College of William and Mary, PO Box 1346, Gloucester Point, Virginia 23062, USA. [4] Marine and Coastal Systems, Deltares, Delft NL-2629 HD, The Netherlands. [5] Centre for Adaptation to a Changing Environment (ACE), Department of Environmental Sciences, Institute of Integrative Biology, ETH Zurich, Zurich 8092, Switzerland. [6] Institut des Sciences de l'Evolution, Université de Montpellier, CNRS, IRD, EPHE, CC065, 34095 Montpellier Cedex 05, France. [7] Aquatic Ecology and Water Quality Management Group, Environmental Science Department, Wageningen University, Wageningen NL-6700 AA, The Netherlands. [8] Patuxent Wildlife Research Center, U.S. Geological Survey (USGS), Duluth, Maryland 20708, USA. Correspondence and requests for materials should be addressed to J.v.B. (email: jim.van.belzen@nioz.nl).

In the current context of global change, understanding ecosystem response to perturbations has become an urgent necessity[1]. Some ecological systems may be resilient to changes while others have been shown to exhibit tipping points caused by the presence of alternative stable states[2]. These systems can catastrophically shift from one state to another in response to sometimes only slight changes in conditions or small perturbations[2–4]. Human dependence on the services and functions these ecosystems provide[1] impels a need to develop methods to forecast their responses to changing conditions and perturbations[2,5]. Mathematical modelling has revealed that generic indicators may exist for a broad class of systems and that can serve as early warnings to inform whether resilience is in decline[2,6]. These indicators are based on the phenomenon of 'critical slowing down', which means that the time needed for a system to recover from a disturbance lengthens when the level of stress applied on the system increases[3,4]. Temporal and spatial statistical signatures of slowing down have been inferred indirectly from fluctuations and correlations in system states[7,8], and highly controlled experiments have provided support that the phenomenon exists in real living systems[7–12]. However, direct measurements of the recovery rates that test this theory in the complex setting of real-world ecosystems, in which sources of heterogeneity and stochasticity are ubiquitous and may obscure signals of looming breakdown[13,14], remain scarce. Hence, our understanding of early warning signals in real-world systems is still insufficient, which severely limits its application to policy decisions and ecosystem management.

Here we bridge this gap between theory and application by examining if critical slowing down can be observed along gradients in environmental stress in tidal marsh ecosystems. Tidal marshes are globally distributed intertidal ecosystems inhabiting energetic and stressful coastal environments at the interface between land and sea[15] (Fig. 1a and Supplementary Fig. 1a). Tidal marshes are amongst the most valuable ecosystems on earth, yet vulnerable to the direct exposure and effects of sea-level rise[16,17]. Field and modelling studies have advanced our understanding of the key drivers and processes governing adaptability and vulnerability of tidal marshes to climate change and sea-level rise[17]. Still, our ability to predict their response remains limited due to inherently nonlinear behaviour resulting from strong biophysical feedback[17,18]. Coupling between vegetation growth, hydrodynamics and soil accretion creates a strong positive feedback that drives wetland formation[17–20] (Fig. 1b). This critical feedback becomes more important for marsh stability as one moves from the more elevated landward part to the lower seaward edge of the marsh, which represents a gradient in inundation stress imposed by tidal seawater. At the same time, these feedbacks create the potential for catastrophic shifts to a bare tidal flat state, where all vegetation and associated functions are lost[18,20,21]. Hence, being able to probe the fragility of these valuable wetlands to catastrophic shifts is particularly pressing because projected sea-level rise and demographic changes in coastal regions can invoke irreversible losses when tipping points are surpassed[17].

In the current study, we analysed long-term (∼30 years) observational data and conducted disturbance-recovery experiments to test if a decline in tidal marsh resilience along a gradient of inundation stress can be observed. The results of both analyses demonstrate that vegetation recovery slows down when inundation by seawater increases, thereby signalling a decline of resilience and corroborating theoretical predictions. In addition, the results reveal that direct assessments of the resilience by measuring recovery rate is much more sensitive to changes in resilience than the assessment based on statistical resilience indicators. These findings suggest that an increased risk of tidal

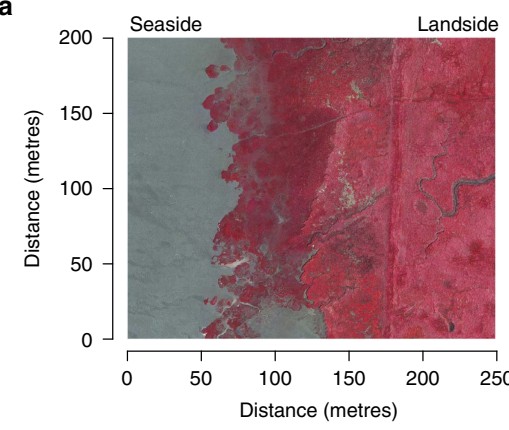

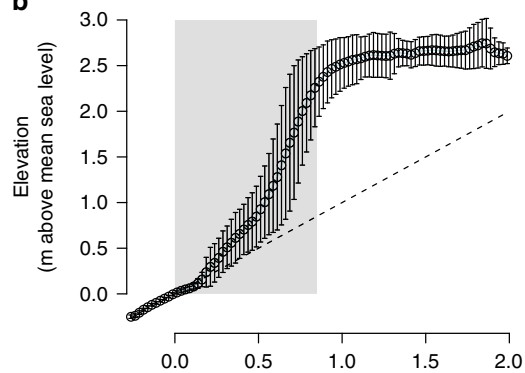

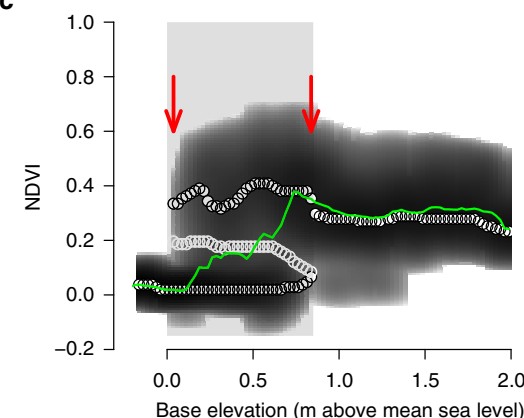

**Figure 1 | Tidal marsh vegetation and elevation along a gradient of seawater inundation.** (**a**) False colour image of tidal marsh vegetation along inundation gradient (red indicates vegetated area) from the sea to land side. (**b**) Cross-shore height profile data show that elevation topography in tidal marshes is shaped by the presence of vegetation. Due to the feedback between vegetation and accumulation of silt and clay the actual elevation (open dots) starts to deviate from the unvegetated base elevation (dashed black line) once vegetation is present on the tidal flat. (**c**) These feedbacks create the risk for catastrophic shifts as suggested by reconstruction of the potential (dark grey shading) of the vegetation along the inundation gradient based on the NDVI. The reconstruction indicates a region of bimodality (grey boxes) at intermediate inundation stress (base elevation) between the high NDVI (biomass) tidal marsh state and a low NDVI (biomass) tidal flat state, highlighting the likely presence of a tipping point in this system (red arrows). White filled and open dots depict these local minima and maxima, respectively. The green line indicates mean NDVI. Error bars indicate s.d. All panels are based on data from site 1 'Hellegat'.

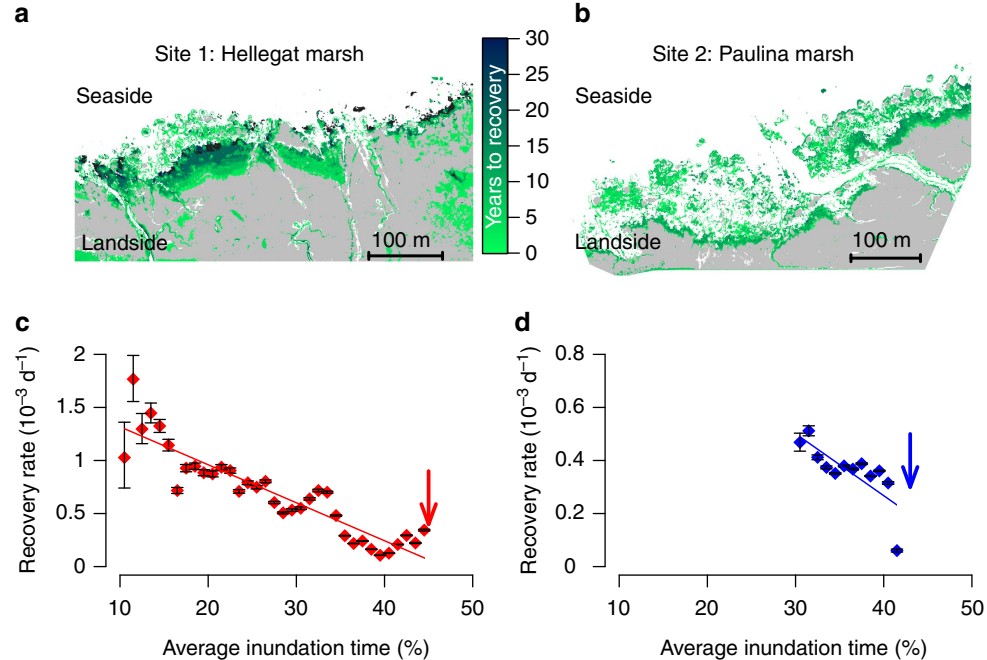

**Figure 2 | Remotely sensed vegetation recovery along a gradient of seawater inundation.** (**a,b**) The spatial distribution of the average time (in years) needed for vegetation to recovery after a perturbation in two Dutch tidal marsh sites. (**c,d**) Estimated recovery rates as a function of the average (multi-annual) inundation time revealing critical slowing down of vegetation recovery with increasing inundation time. Grey area (in **a**, **b**) represents never disturbed stable marsh vegetation, white area is the unvegetated tidal flat. Red symbols (**c**) depict recovery rates at site 1 'Hellegat' (NL), blue symbols (**d**) are rates at site 2 'Paulina' (NL) and error bars depict c.i. Arrows depict the critical condition of marsh vegetation. Linear regression: site 1, $R^2 = 0.83$, $P < 0.001$; site 2, $R^2 = 0.59$, $P < 0.005$.

marsh collapse, for instance in response to intensified inundation due to sea-level rise, can be assessed if a further slowing down of vegetation recovery is observed.

## Results

**Critical slowing down in remotely sensed imagery**. We established direct support for the presence of tipping points and critical slowing down along an inundation gradient by analysing time series of aerial images of tidal marsh development and recovery from episodic erosion in two marsh sites in The Netherlands. Aerial images revealed bimodality at intermediate inundation stress (intertidal elevation) between the high biomass tidal marsh state and a low biomass tidal flat state[22–24] (site 1: Fig. 1c, site 2: Supplementary Fig. 1b). The critical condition at which the two tidal marshes are observed to tip from high-to-low biomass are found at base elevation 0.0 and 0.3 m above mean sea level (Fig. 1c and Supplementary Fig. 1b, left arrows), which corresponds with an inundation time of 47% and 43%, respectively. In combination with experimental and observational studies revealing clear density-dependent thresholds for vegetation establishment in these same tidal marshes (refs 20,21,25), our results highlight the potential presence of bistability and tipping points at the two tidal marshes under study.

To test for the presence of critical slowing down along the tidal inundation gradient, we determined whether temporal changes in local vegetation cover can provide information on the time needed for vegetation to recover from local disturbances (Fig. 2a,b), as a function of the location along the inundation gradient (Fig. 2c,d). In line with our theoretical predictions (Supplementary Note 1 and Supplementary Fig. 2c), recovery rates decreased with increasing inundation stress along the elevation gradient from high-to-low marsh (Fig. 2c,d and

Supplementary Table 2). Hence, our observations corroborate the prediction that the stress imposed by seawater inundation slowed down recovery, and thereby impairs the resilience of tidal marshes.

**Critical slowing down in experimental disturbances**. We then experimentally tested our hypothesis that recovery of marsh vegetation slowed down with increasing inundation by disturbing vegetation at both the two Dutch sites used in the time-series analysis, as well as in a North American marsh. The two Dutch field sites are macrotidal, polyhaline and the pioneer vegetation is dominated by the cordgrass *Spartina anglica*. Here, mineral deposition drives soil elevation change and as a consequence, vegetated marshes can keep pace relatively easily with sea-level rise[26]. To test the generality of our results to globally diverse and important marshes, we conducted a similar disturbance-recovery experiment (Methods) at a microtidal, mesohaline tidal marsh in North America that is dominated by the bulrush *Schoenoplectus americanus*. In this marsh, root growth and subsequent belowground biomass accumulation are the main drivers of slow intertidal elevation change, forming organic rich soils that are rapidly submerging in response to sea-level rise[27]. In all sites, in early summer, small tussocks of tidal marsh vegetation were transplanted to different intertidal elevations, after which vegetation was disturbed by clipping all aboveground biomass (Methods). Recovery was measured at the end of the growing season by comparing biomass regrowth of the clipped tussocks with that of unclipped transplanted tussocks (Methods, Supplementary Note 2).

Consistent with critical slowing down, the disturbance-recovery experiments revealed unambiguously that recovery rates declined with increasing submergence in disparate marshes on both sides of the Atlantic Ocean (Fig. 3). Although the threshold

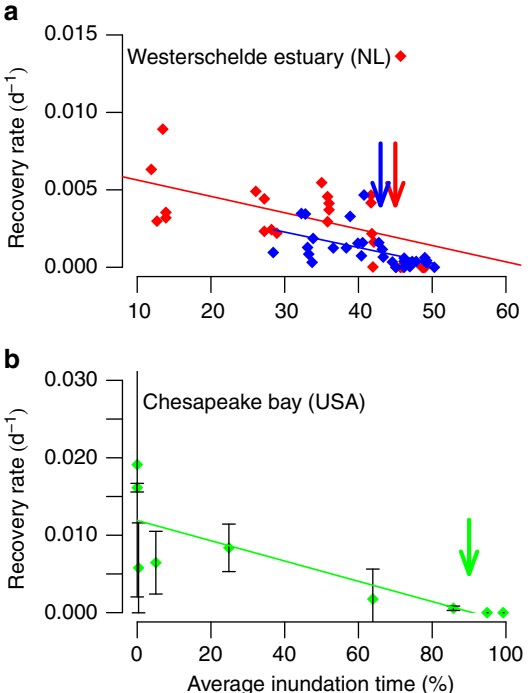

**Figure 3 | Experimental recovery rates of tidal marsh vegetation.**
(**a**) Biomass recovery after mowing (disturbance) within one growing
season in two Dutch tidal marsh sites. (**b**) Recovery after mowing within
one growing season in the North American experiment. Red symbols
indicate measurements at site 1 'Hellegat' (NL), blue symbols are from site
2 'Paulina' (NL), and green symbols site 3 'Blackwater' (USA). Arrows
depict the critical condition of marsh vegetation. (Linear regression: site 1,
$R^2 = 0.18$, $P = 0.02$; site 2, $R^2 = 0.28$, $P < 0.005$; US site 3, $R^2 = 0.66$,
$P < 0.01$.) Error bars depict s.e.m.

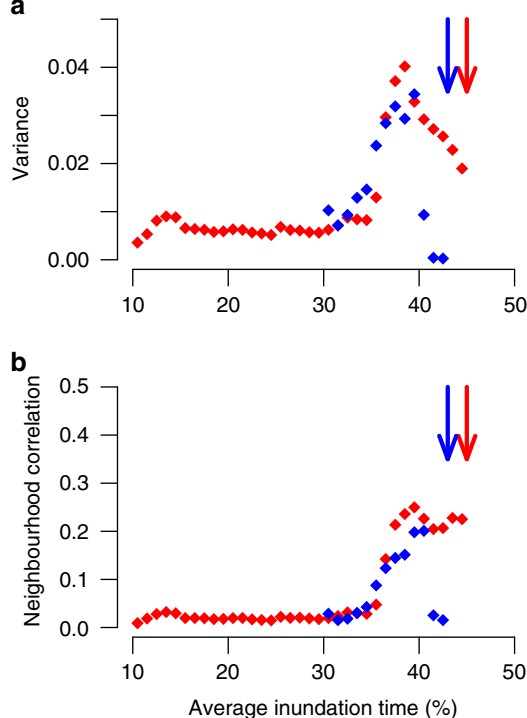

**Figure 4 | Remotely sensed indirect resilience indicators along a gradient
of seawater inundation.** (**a**) Variance and (**b**) neighbourhood correlation of
aerial images as a function of the underlying stress gradient by seawater
inundation for which only one of two sites show a significant trend with
increasing inundation stress. Red symbols indicate site 1 'Hellegat' (NL),
blue symbols depict site 2 'Paulina' (NL). Arrows depict the critical
condition of marsh vegetation. Linear regression: (**a**) site 1, $R^2 = 0.51$,
$P < 0.001$; site 2, $R^2 = 0.00$, $P = 0.96$, (**b**) site 1: $R^2 = 0.59$, $P < 0.001$; site 2:
$R^2 = 0.20$, $P = 0.13$.

inundation time at which tidal marsh vegetation ceased to survive
was very different between the Dutch (Fig. 3a) and North
American (Fig. 3b) sites, the response trend was strikingly similar.
Differences in, for example, tidal range, exposure to waves,
salinity and species physiological tolerances to stress likely explain
the difference in threshold levels. Due to the microtidal range of
the Northern American marsh site ($< 25$ cm), the vegetation is
rarely submerged completely, allowing it to survive longer
inundation times up to $\sim 90\%$ (ref. 28). Still, these results
support the idea that critical slowing down is a generic
phenomenon that can be used to assess the resilience of diverse
marsh ecosystems around the world under varying hydrodynamic
and geomorphological conditions. Hence, these differences
between the Dutch and North American marsh sites stipulate
the need for critical slowing down as alternative indicator to
assess site-specific differences in resilience against seawater
flooding.

**Testing indirect spatial signatures of critical slowing down.**
Finally, we tested if critical slowing down could be detected in our
data indirectly from spatial statistics. Theoretical studies have
predicted that spatial variance and autocorrelation could provide
an alternative, less data demanding, fingerprint of declining
resilience[2,7,8,29]. Especially, when direct measurements on critical
slowing down will not be feasible due to lack of long-term data set
or inaccessibility of the area of interest, these indicators could be
more convenient. In particular, spatial variance is predicted to
rise and neighbouring sites become more alike resulting in an

increased correlation, as a consequence of a slower response to
disturbance when stressed[29]. To test if these trends can be
detected in our data, we calculated how spatial variance and
correlation with neighbouring sites changed along the inundation
gradient of the two Dutch sites (Methods).

We observed that trends in the indirect resilience indicators
were not consistently significant to forecast an impending
catastrophic shift. Even though a sharp increase in both resilience
indicators (that is, spatial variance and correlation with
neighbouring sites) near the tipping point can be observed
for both sites (Fig. 4 and Supplementary Table 2), the trends in
resilience indicators showed to be significant only in one site
(site 1, 'Hellegat'). The resilience indicators failed in the other
site (site 2, 'Paulina'). These results were independent of the
spatial resolution at which these indirect resilience indicators
were analysed (Supplementary Table 3). Overall, the indirect
resilience indicators were less sensitive to changes in inundation
stress compared to the direct measurements of recovery rates, as
evidenced by the lower correlation between the indicators and the
average inundation time (Supplementary Table 2). This difference
in correlation strength can be explained by the difference in
response along the inundation gradient: Recovery rates slowed
down linearly along the whole gradient (Fig. 2c,d), while the
indirect resilience indicators show little response before 32%
inundation time after which they rapidly increased (Fig. 4). Thus,
our results highlight the superiority of direct measurement of
critical slowing down from aerial observations or manipulative

experiments over indirect statistical measures such as spatial variance and autocorrelation.

## Discussion

Anthropogenic and climate stressors are changing and degrading ecosystems worldwide at alarming rates[1], emphasizing the need to identify indicators of resilience loss. This is particularly true for coastal wetlands, which are globally threatened by sea-level rise and growing coastal populations, but also desired for coastal protection[17,30], as nursery grounds for commercially important fisheries, as biofilters and for carbon sequestration[16,17]. In wetlands and other ecosystems, thresholds and nonlinearity make precise predictions of their response to changing environmental conditions difficult. This complexity makes it challenging to identify when management intervention is needed to prevent a dramatic change in ecosystem state[1,9,31]. Our results provide clear support that the phenomenon of critical slowing down can be used to identify declining resilience in natural systems, despite many sources of heterogeneity and stochasticity. Therefore, the application of critical slowing down to evaluate increased vulnerability of ecosystems to changing conditions—such as sea level in the case of tidal marshes—will allow for timely adoption of appropriate management strategies before catastrophic and irreversible loss of functioning occurs.

Our results highlight the superiority of direct measurements of recovery rates over indirect resilience indicators, such as spatial variance and correlation with neighbouring sites. The inconsistency between sites in revealing declining resilience along an inundation gradient suggests that indirect statistical signals can be too unreliable to provide a dependable signal of the loss of resilience. The main reason for this is that the performance of indirect statistical indicators, such as variance and correlation between neighbouring sites, depends on specific assumptions that are difficult to meet outside the territories of artificial or controlled systems[13,14,32]. Specifically, statistical signs of upcoming catastrophic transitions may fail when variability is relatively high due to internal feedback processes, external stochastic forcing, or other sources of heterogeneity[13,14,32]. Nevertheless, with the rapid development of satellite systems, direct measurements of recovery rates become feasible at large spatial and temporal scales. Consequently, quantifying critical slowing down using remotely sensed images, as done in our study with aerial photographs, comes within reach for a growing number of ecosystems around the globe, and will provide a valuable tool for informed ecosystem management[2,23].

## Methods

**Study sites.** We focused our remote-sensing analysis and corresponding field experiments in the Netherlands on two polyhaline tidal marsh sites in the Westerschelde Estuary for which over 25 years of tidal marsh development could be established based on aerial photographs (site 1, 'Hellegat', 51.367°N, 3.95°E; site 2, 'Paulina', 51.355°N, 3.715°E). This turbid and well-mixed estuary is macrotidal with 3.8 m mean tidal range at the mouth of the estuary near Vlissingen and about 5.0 m located 80 km upstream near Antwerp[33]. The sites experience a spring tidal range of 4.86 and 4.7 m, respectively[26]. The cordgrass *Spartina anglica* is the prime colonizer in this area and dominates the lower tidal marsh, but is successively replaced by sedges (*Aster tripolium, Limonium vulgare, Suaeda maritima* and *Plantago maritima*) and grasses (*Puccinellia maritime* and *Elymus arenarius*) at higher elevations. Soil accretion in these marshes is largely driven by mineralogical depositon[25,26].

To test the generality of our results, we conducted a parallel experiment in a large brackish marsh on the Atlantic Coast of North America. The study site is adjacent to the Blackwater River, a tributary of the Chesapeake Bay (Maryland, USA) (site 3, 'Blackwater', 38.40°N, 76.07°W). Changes in water level are primarily driven by meteorological events, with mean astronomical tides of < 0.25 m. Long-term porewater salinities average 10 p.p.t. within the marsh soil, and intertidal vegetation is dominated by *Schoenoplectus americanus* and *Spartina*

*patens*[27]. The site receives little mineral sediment from the catchment, and has lost ~50% of its marshland since the 1930s (refs 34,35). Thus, the US study site differs fundamentally from the Netherlands study sites in tidal range, salinity, vegetation, sediment supply and historical stability.

**Remotely sensed spatial data.** Development of tidal marshes in the Dutch study area is known from aerial photo surveys conducted systematically since the 1970s by Rijkswaterstaat. Time series of false colour aerial images (near infrared (NIR), red (R) and green (G) colour bands) over 25 years were available for two sites (see Supplementary Table 1 for which years were available and used per site). These photos were digitized at 0.25 m spatial resolution and classified for tidal marsh vegetation presence and absence based on supervised classification[33].

**Reconstruction of basins of attraction.** We tested whether tidal marshes responded gradually or abruptly to changes in inundation by reconstructing the basins of attraction along the inundation gradient[22–24]. The reconstruction was based on Normalized Difference Vegetation Index (NDVI) values calculated from false colour aerial images in 2010 as $NDVI = (NIR-R)/(NIR+R)$ (data file v_Westerschelde_2010.ecw from Rijkswaterstaat http://www.rijkswaterstaat.nl/apps/geoservices/geodata/dmc/orthofotomozaieken_ecologie/geogegevens/raster/). NDVI values were binned based on inundation ranges after which the probability density function $P_d$ was estimated. The vegetation potential $U$ is directly related to the density function and calculated as $U = -\log(P_d)$. A 3 by 3 median filter was used to smooth out small irregularities in the reconstructed potential. Local minima and maxima in the reconstructed potential landscape were interpreted as the basins of attraction and repulsion, respectively.

**Remotely sensed disturbance and recovery of tidal marshes.** The sequential data of presence and absence of tidal marshes vegetation was treated as a natural disturbance-recovery experiment[36]. Disturbance of vegetation was detected if vegetation present on the classified aerial images was absent in the subsequent image. By recording the time needed for a grid cell to recover, once disturbed, a record of vegetation recovery time was established on a pixel-by-pixel basis. As we assumed that inundation duration is the main stressor of the tidal marsh ecosystem, we binned the obtained recovery times based on the local average inundation time (in %), before estimation of the recovery rates. The bin width was 1% (which corresponds with 0.12 h per tide). Inundation frequency maps were calculated from bathymetry maps and actual recorded water level measurements at a nearby water level gauge (at Terneuzen; by Rijkswaterstaat http://live.waterbase.nl). Recovery rates were estimated, for each bin, from the established recovery times using a maximum likelihood estimator for the exponential model. To avoid overestimation of the recovery rates the maximum likelihood estimator corrects for censored recovery times (grids cells that were disturbed but did not recover by the end of the time series). A full protocol for the timing of recovery based on sequential spatial data can be found in ref. 36. We focused our analysis on the areas in the Dutch sites where the disturbance-recovery experiments were executed.

**Disturbance-recovery experiments.** Manipulative field experiments[37] were conducted in Europe and North America to verify if critical slowing down responses could be detected in a field setting. These field experiments tested for thresholds of vegetation tolerance to flooding. At the Dutch study sites randomized tussocks of *Spartina anglica* vegetation were transplanted to six levels along the inundation gradient. Because the present-day marsh edge does not necessarily reflect the position of the actual extinction threshold for vegetation tolerance in this actively evolving system, the six inundation levels were chosen in such a way that they encompass both the leading marsh edge and the extinction threshold. Transplanted vegetation tussocks were approximately 0.0625 m$^2$ (0.25 by 0.25 m) and included roots and soil to a depth of 0.4–0.5 m (site 1: 1,497 ± 420 stems m$^{-2}$; site 2: 721 ± 304 stems m$^{-2}$). At every level, vegetation exposed to mowing disturbances is compared to transplanted control tussocks to measure the relative recovery. The setup of control and disturbed tussocks was replicated five times at every inundation level. The experiment ran for about 4 months before harvesting all aboveground biomass (site 1: from 20 May, when the disturbance was applied, to 1 September 2011 when both disturbed and control plots were harvested (104 days); site 2: from 13 July to 20 October 2011 (129 days)). After setup, the intertidal height of the experimental plots was measured using a dGPS (Leica Geosystems) to determine the actual inundation time per plot.

At our US study site, we measured the response of plants to disturbance across a gradient in inundation times by transplanting tussocks of *Schoenoplectus americanus* into mesocosms of different elevation[37]. The mesocosms were arranged into structures commonly described as 'marsh organs'. Here we utilize two marsh organs each containing 54 mesocosms constructed of 6-inch diameter (0.0182 m$^2$) polyvinyl chloride pipe, arranged into nine rows containing six pipes of identical elevation. The experimental design and basic patterns of vegetation response to inundation have been previously described[27]. Both organs were planted on 11 April 2012 and harvested on 21 August 2012. Plants in one organ were disturbed by

clipping all aboveground vegetation on 20 June 2012 (62 days), and left undisturbed in a control organ.

In each experiment, plant stems were clipped at the soil surface to isolate standing aboveground material, and sorted into live and dead fractions by species. Aboveground biomass was washed and dried at 70 °C to a constant weight. Recovery rates are estimated assuming exponential distribution[4] as $\lambda = -\log(1-f)/\Delta t$ (see Supplementary Note 2 and ref. 37), in which the relative recovery $f$ is the fraction of total aboveground biomass recovered comparative to the undisturbed transplanted controls and $\Delta t$ is the duration of the experiment after which recovery was measured. We used Pearson product–moment correlation coefficient to quantify the strength of the trend along the inundation gradient of the resilience indicators measured[38]. The choice for Pearson product–moment correlation coefficient over the Kendall's $\tau$ did not affect our results in a qualitative way.

A full protocol for the disturbance-recovery experiments can be found in ref. 37.

**Indirect spatial indicators of resilience.** Using the same binning scheme as used for the direct measurement of recovery rate (that is, resilience) from the remotely sensed vegetation data we measured a range of spatial statistics, following ref. 38. We measured coverage, and the regular proposed measures of resilience variance and neighbourhood correlation (that is, spatial autocorrelation with the four neighbouring pixels) for the spatial data in each of the NDVI maps[2,7–9]. We furthermore checked the sensitivity of these analysis for the resolution of the data by coarse graining the NDVI maps to a spatial resolution of 1, 5 and 10 m.

**Data availability.** The aerial images, experimental data and the code used to analyse the data and model tidal marsh dynamics that support the findings of this study are available from the corresponding author upon reasonable request and via Dryad Digital Repository: http://dx.doi.org/10.5061/dryad.7174h.

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

## Acknowledgements

We thank B.R. Silliman for discussion on the experimental setup. A. Jacobs, J. van Dalen and L. van IJzerloo are recognized for their assistance in the field and A. Wielemaker-van den Dool for assistance with GIS. We thank VFA. de Witte for useful comments on earlier versions of the manuscript. The work of J.v.B and T.J.B. is funded by the European Commission through FP7 ENV2009-1, Contract 244104-THESEUS. J.v.B. was further supported by the VNSC project 'Vegetation modelling HPP' (contract 3109 1805) and the Schure-Beijerinck-Popping fund of the Royal Dutch Academy of Science to visit S.K. and V.D. T.J.B. was further supported by the NWO funded BE-SAFE project grant 850.13.011. G.R.G. and M.L.K. were supported by funds from the US Geological Survey Climate and Land Use Research & Development program, and the US National Science Foundation LTER 1237733. Any use of trade, product, or firm names is for descriptive purposes only and does not imply endorsement by the US Government.

## Author contributions

J.v.B., D.v.d.W., P.M.J.H., V.D., S.K. and M.S. analysed the remotely sensed data. J.v.B., J.v.d.K. and T.J.B. designed and conceived the Dutch experiments, which were performed by J.v.B. M.L.K. and G.R.G. conceived and performed all North American experiments. J.v.B. and M.L.K. analysed all field data. J.v.B., J.v.d.K., M.L.K. and T.J.B. wrote the paper. All authors discussed the results and commented on the manuscript.

## Additional information

**Competing interests:** The authors declare no competing financial interests.

