## [Peer Review File · Nature Communications]

Reviewers' Comments:

Reviewer #1 (Remarks to the Author):

The paper is a resubmission of “Direct evidence for critical slowing down when approaching tipping points in tidal marshes“ after revision. All my comments on the first version of the paper have been addressed and the paper has largely been improved. However, while the work is actually very interesting, one major concern still remains.

No critical condition (i.e., critical recovery rate), can be identified as an indicator (i.e., an early warning signal) of impending tipping point. Data plotted in Figs 2c,d and 3a,b show that the recovery rate approximately decreases, at a constant rate, with inundation time increasing, and no particular behavior forewarns when conditions are close to the tipping point. Sorry, I don't see how one can predict the location of tipping point by measuring (well before “catastrophic loss of functioning imposes irreversible damage”) the slowing down of recovery rate.

In addition,

Lines 139-142. You state “Both variance and correlation with neighbours showed a significant trend in one site (site 1, ‘Hellegat’), but failed in the other site (site 2, ‘Paulina’) to forecast an impending catastrophic shift (Fig. 4 and Supplementary Table 2)”. This is not much different from your statement of the previous version “Correlation with neighbours showed a significant trend in one site, but failed in the other site to forecast an impending catastrophic shift (Fig. 4 and Extended Data Table 2)”. However, Fig. 4 of the original paper is dramatically different from the new Fig. 4. Importantly, Fig. 4 of the original paper actually shows that i) correlation with neighbours is significant in site 1 but not in site 2 and ii) no particular behavior can be detected as an early warning signal for impending tipping point. On the contrary, the new Fig. 4 shows that both variance and correlation with neighbours sharply increase, in both sites, just before the tipping point, i.e., they look as effective indicators of an impending catastrophic shift.

Minor points

Fig. 1c shows the NDVI as a function of base elevation, whereas, Supplementary Fig. 1b shows the NDVI as a function of the average inundation time. I suggest using both scales in Fig. 1c line 425. Unvegetated misspelled

Reviewer #2 (Remarks to the Author):

This is an excellent work. I recommend the manuscript be accepted pending minor changes. Specifically,

1. Some choices of language are odd and/or unnecessarily wordy. For instance
 - "demarking" is an odd verb for characterizing tipping points (line 77)
 - "exploring time series" is a strange way to describe statistical analysis (line 79)
 - suggest "determined whether temporal changes" (line 92)
 - suggest to substitute "prediction" for "expectation" (line 98)
 - use of word "Merely" inappropriate (line 129)
 - suggest to substitute "detected" for "picked up" (line 133)
 - antecedent of pronoun "this" is unclear (line 140, again in line 174)
 - "hamper making precise predictions" is an awkward construction (line 169)
2. Statements in lines 157-161 would be more appropriate in discussion than results
3. Saying that the evidence for CSD is "much needed" is rather more subjective than necessary and perhaps oversells this line of work (line 172); why not just say that this study provides evidence of this phenomenon?

John Drake

Appendix: Response to the reviewers of *Nature*

Note by authors: For clarity, we wrote our answers in blue between the original review report of 3 referees. Because some referees raised the same issues we numbered the answers to allow easy reference between answers.

Referee #1 (Remarks to the Author):

Comments on the paper, "Direct evidence for critical slowing down when approaching tipping points in tidal marshes" by Jim van Belzen, Johan van de Koppel, Matthew L. Kirwan, Daphne van der Wal, Peter M.J. Herman, Vasilis Dakos, Sonia Kéfi, Marten Scheffer, Glenn R. Guntenspergen and Tjeerd J. Bouma. [2]

The aim of this work is to provide empirical support for the "critical slowing down" theory through observational and experimental tests on tidal marsh ecosystems. The issue is surely of interest to the journal's readership. However, given my comments below, I think that the paper is not suitable for publication in *Nature*. It is well known that, e.g., the fate of vegetation immersed in a hostile environment is either to grow thin or to die (this is a common belief based on everyday observation). The big question is "how can I determine whether an environment is hostile?". The answer to the question can be "critical slowing down", i.e., an environment is hostile if the rate of recovery following disturbance goes below a given threshold. To prove the latter one must show that conditions (tipping point) discriminating the fate (here, of marsh vegetation) actually exist. In addition, for the approach to be effective, these threshold conditions must be identified and quantified. [2]

If I see, no critical condition (i.e., no critical recovery rate), which can be used as an early warning signal, is identified in the paper. Data plotted in Figures 2c,d and 3a,b suggest that the recovery rate decreases at a constant rate with inundation time increasing; accordingly, neither the recovery rate nor the inundation time can be used as an early warning signal of extinction, since a threshold (critical recovery rate or inundation time) is lacking. [2]

#A1, answer by authors: We agree with referee #1 that we did not clearly define the critical conditions at which the tipping points are found. Nevertheless, we have a clear idea about the critical conditions due to the potential analysis shown in Fig 1c and Supplementary Fig 1b we presented in our study. Therefore, we now clearly demarcate the position of the tipping points (the critical conditions) in Figures 2-4 based on these results. The tipping point of interest was defined as the lowest intertidal elevation along the environmental gradient (i.e. inundation time gradient) at which two local minima are found, just before switching to one minima of low biomass (NDVI). Although two switches can be identified per gradient representing tipping points (i.e. in the case of a system consisting of two alternate stable states) we focused on the switch (tipping point) from high to low biomass (i.e. NDVI).

Furthermore, we now explicitly mention these critical conditions in the text, from line 80-89:

“Aerial images revealed bimodality at intermediate inundation stress (intertidal elevation) between the high biomass tidal marsh state and a low biomass tidal flat state²²⁻²⁴ (Site 1: Fig. 1c, Site 2: Supplementary Fig. 1b). The critical conditions at which the tidal marshes tip from high to low biomass are found at base elevation 0.1 and 0.8 m above mean sea level, which respectively corresponds with 45% and 43% inundation time respectively. “

Also, the chosen indicator, or proxy, for the recovery rate, i.e., the inundation period, does not look effective in determining whether the system is close to the tipping point or not. Very different inundation periods at which tidal marsh vegetation ceased to survive are found for the Dutch and North American sites.

The Authors correctly state that the different recovery rates measured in the two sites can be explained by the "differences in e.g., tidal range, exposure to waves, salinity, and species physiological tolerances to stress", etc. This confirms that indicators discriminating the fate of marsh vegetation are not identified yet. Importantly, the question "which is the threshold (critical) recovery rate, e.g., for cordgrass?" remains unanswered.□□□□

#A2, answer by authors: These comments made it clear to us that we did not explain sufficiently how we identified the critical conditions under which tipping points occur in these tidal marshes, both for the Netherlands and for the US. We therefore added the following clarifications to the text, from line 121-130:

“Differences in e.g. tidal range, exposure to waves, salinity, and species physiological tolerances to stress likely explain the difference in threshold levels. Due to the micro tidal range of the Northern American marsh site (<25cm), the vegetation is rarely submerged completely, allowing it to survive longer inundation times up to about 90%²⁸. Still, these results support the idea that critical slowing down is a generic phenomenon that can be used to assess the resilience of diverse marsh ecosystems around the world under varying hydrodynamic and geomorphological conditions. Merely, these differences between the Dutch and North American marsh sites stipulates the need for alternative indicators to assess site specific differences in resilience against seawater flooding. “

Referee #2 (Remarks to the Author):

Ecological systems commonly exhibit tipping points at which the dynamics change qualitatively. Methods for anticipating these tipping points and measuring resilience are an important area of research. While much theory has been developed and there are some experimental demonstrations, there have been few field trials to demonstrate the phenomenon of critical slowing down in nature or field based evaluations of candidate early warning signals. The submitted work seeks to fill this gap with a combined experimental-theoretical

project investigating bistable marsh vegetation. The study is well chosen for this purpose and was expertly performed. Although I have minor comments about the presentation of work, I have no concerns about the intellectual rigor. Therefore, my negative recommendation to Nature reflects only my concern about the noteworthiness of the findings and comprehensiveness of the study (elaborated below), not criticism of the work as presented.☐☐

#A3, answer by authors: We are very grateful for the recognition of the intellectual rigor of our study. Given the global-scale threat imposed on valuable ecosystems by global change processes, especially in coastal zones, we do not agree to the lack of “noteworthiness and comprehensiveness”. This is also not supported by the comments of the other reviewers. We have apparently not stressed this sufficiently. We now stipulate the importance of our work in line 62-74:

“Still, our ability to predict their response remains limited due to inherently nonlinear behaviour resulting from strong biophysical feedback^{17,18}. Coupling between vegetation growth, hydrodynamics and soil accretion creates a strong positive feedback that drives wetland formation¹⁷⁻²⁰ (Fig. 1b). [...] At the same time, these feedbacks create the potential for catastrophic shifts to a bare tidal flat state, where all vegetation and associated functions are lost^{18,20,21}. Hence, being able to probe the fragility of these valuable wetlands to catastrophic shifts is particularly pressing because projected sea level rise and demographic changes in coastal regions can invoke irreversible losses when tipping points are surpassed¹⁷.”

The study's key finding is that critical slowing down is observed to increase with an environmental stressor. Critical slowing down, in this case, is the instantaneous rate of growth of vegetation. That vegetation grows slower when exposed to an environmental stressor (eg inundation) is unobjectionable, but also not noteworthy. The authors have not quantified the location of the tipping point, therefore no quantitative statements can be made about the change in instantaneous growth rate with respect to proximity to the tipping point.☐

#A4, answer by authors: We agree with referee #2 that the identification of the critical conditions at which the tipping points are found, was not clearly explained in the manuscript. However, the analysis and experiments we performed did allow us to identify these critical conditions and solved this issue by clearly demarking the position of the critical conditions in the text and figures. For the detailed explanation, please see answer #A1 in the response to reviewer 1, whom had raised exactly the same concern.

The secondary finding is that theoretical statistics predicted to coincide with critical slowing down (eg increase in autocorrelation) were not consistently observed. Of course, it is useful to know when a theoretically predicted phenomenon is too fragile to observe in nature. But, the problem here is that we don't really know why it hasn't been observed. From the analysis performed, one can only speculate. One opportunity for exploring this idea further is to better

develop the theoretical model presented in the supplementary material. Here, the authors have developed a system that exhibits the anticipated behavior. Can they also continue to complicate this system until the behavior disappears? Why does it disappear? How much heterogeneity is required to corrupt the signal? These are theoretically tractable questions. In my view, if the secondary finding of this paper (the signal can't be detected in nature) is to warrant publication in Nature, we must know why it hasn't been found, not just that it wasn't found on this occasion.☒☒

#A5, answer by authors: We agree that we didn't show in depth why the spatial indicators are not working in this particular system. Yet, we did look at the performance of various spatial indicators and at different scales by coarse graining the spatial data, but did not report on these results to keep the manuscript as concise as possible and perhaps use them for another future paper. Given this question by the referee we see the importance of including these data within this manuscript. Hence, we now added these findings as Supplementary Table 3.

These findings further support our secondary conclusion that the indirect spatial indicators are not showing consistent results between sites. We did not extend the models analysis to see at what levels of stochastic forcing the spatial early warning signals break down, as the main objective of our paper was to come up with the greatly lacking observational data rather than adding more model analysis on the limitations of early warning signals (see e.g. Dakos et al 2012 [ref 14]).

We added a new section "Indirect spatial indicators" from line 292-299 to the methods to explain how spatial indicators and course-graining was implemented.

Furthermore, we added our findings to the manuscript at line 142-150:

"Both variance and correlation with neighbours showed a significant trend in one site (site 1, 'Hellegat'), but failed in the other site (site 2, 'Paulina') to forecast an impending catastrophic shift (Fig. 4 and Supplementary Table 2). Moreover, these results were not sensitive to the spatial resolution at which the early warning signals were analysed (Supplementary Table 3). The data from site 2 consistently showed no reliable trend in the resilience indicators along the inundation gradient (Supplementary Table 3). The inconsistency between sites is not entirely surprising because the robustness of indirect statistical indicators such as variance and correlation between neighbouring sites is restricted to specific assumptions that are difficult to meet outside the territories of artificial or controlled systems^{13,14,30}."

Other issues the authors may wish to consider include the following:☒

- The paper emphasized the study's uniqueness as a field study. The obvious comparison is Steve Carpenter's paper "Early warnings of regime shifts: a whole-ecosystem experiment" published in 2011 in Science. For completeness, this paper must be addressed and similarities and differences described. Particularly, the authors should explain why Carpenter et al found evidence of critical slowing down in statistical indicators.

#A6, answer by authors: What sets our work apart is that we show declining resilience directly from the spatiotemporal data by measuring how recovery rates decrease with increasing stress, while other empirical studies infer this indirectly from spatial statistics.

We clarified this by stating more explicitly in the text, at line 46-52:

“Temporal and spatial statistical signatures of slowing down have been inferred indirectly from fluctuations and correlations in systems states^{7,8}, and highly controlled experiments have provided support that the phenomenon exists in real living systems⁷⁻¹². However, direct measurements of the recovery rates that test this theory in the complex setting of real-world ecosystems, in which sources of heterogeneity and stochasticity are ubiquitous and may obscure signals of looming breakdown^{13,14}, remain scarce.”

-Independent empirical validation of bistability in the two experimental systems would be valuable☐

#A7, answer by authors: The absence of independent verification of bistability in our study systems has no impact on our main finding of critical slowing down (CSD). We want to point out that CSD can also be found in systems without bistability (see e.g. Kefi et al 2012 in *Oikos* and Drake and Griffen 2010 in *Nature* for examples of CSD and early warnings in trans-critical systems). Nevertheless, we understand that we pointed at the utility of using CSD as a way to probe apparent discontinuous behavior to environmental stressors, highlighting the proximity to tipping points.

We show, using the potential analysis, that the patterns of local minima (maxima in the frequency distributions) along the environmental gradients in two Dutch tidal marshes are consistent with the presence of bistability (see Fig 1c and Ext Data Fig 1b). Furthermore, we refer to published experimental work done in these same tidal marshes, which points at size/biomass thresholds for successful establishment of tidal marsh vegetation. These two independent lines of evidence make a strong case for the presence of bistability in our tidal marshes.

However, we agree that this point was not stipulated well enough. Therefore, we emphasized and clarified this point in the text at line 85-89:

“In combination with experimental and observational studies revealing clear density-dependent thresholds for vegetation establishment in these same tidal marshes (ref. 20, 21 & 25), our results highlight the potential presence of bistability and tipping points at the two tidal marshes under study.”

- Empirical measurements are made of net primary production for a year, but it is unclear how this integrated variable relates to the instantaneous states described by the theory -- this is a theoretically tractable problem and should be addressed☐☐

#A8, answer by authors: We agree with the reviewer #2 that it is not clear from the manuscript how we calculated recovery rates from our empirical measurements. Therefore, we added a Supplementary Note 2 which gives an in-depth account of the method for inferring recovery rates from net primary production measurements.

In closing, I admire the work performed, particularly the replication (two marshes on two continents!) and hope these comments are helpful in presenting the work most compellingly in the future. ☐☐

Sincerely yours,
John Drake

Referee #3 (Remarks to the Author):

This paper provides strong support for critical slowing down (CSD) prior to catastrophic transition using long-term data from a disturbance gradient in real-world ecosystems. So far there are few tests of CSD theories using field data from ecosystems. Therefore the findings of this paper are important and should be of wide interest.

Because high-quality time series for testing CSD theories are hard to come by, there has been considerable interest in using spatial indicators of resilience as a surrogate. If spatial indicators are sensitive to loss of resilience, then it might be possible to infer resilience from occasional samples of spatial pattern, without the need for high-frequency time series. The authors report that local (short-distance) spatial autocorrelation did not decrease along with CSD, suggesting that this indicator may not be reliable in the field. This finding is surprising because it appears contrary to modeling results (some of which are cited in the manuscript) and one field study where spatial patterns of fish behavior did signal declining resilience (<http://dx.doi.org/10.1890/ES13-00398.1>).

#A9, answer by authors: The above issue was also raised by referee #2. What set our work apart from other empirical studies, is that we show declining resilience directly from the spatiotemporal data by measuring how recovery rates decrease with increasing stress. In contrast, other empirical studies infer a decrease in recovery rates indirectly from statistics such as correlation between two consecutive observations in time, an approach that proved less reliable in our study. How we clarified this point in more detail can be found in answer #A6 in answer to referee #2.

Some of the authors have published on diverse methods for discerning loss of resilience in spatial data (PLoS ONE 9:e92097). The local correlations presented in the manuscript depend on the spatial scale of the data. In contrast, many of the methods that have been explored in other work use many scales (spectra) or properties of frequency distributions to discern changes in resilience. While the result reported in the manuscript is quite interesting, it is limited to a single method at one spatial scale. Perhaps it is too soon to throw out spatial approaches, and instead we should be looking at a wider range of statistical methods for studying the signals of resilience in spatial pattern.

#A10, answer by authors: We agree that we didn't show why the spatial indicators are not working in this particular system. This point was also raised by referee #2. In the revised version, we did look at the performance of various spatial indicators and added this information to the extended data. See for a more detailed answer how we resolved this issue answer #A5 to referee #2.

Minor points:

Line 100 Schoenoplectus misspelled

#A11, answer by authors: spelling correct

Ref 27 what is the year?

#A12, answer by authors: corrected to 2011